# Long-term memory is formed immediately without the need for protein synthesis-dependent consolidation in *Drosophila*

Bohan Zhao [1,2], Jiameng Sun[1,2], Xuchen Zhang[1], Han Mo[1], Yijun Niu[1], Qian Li[1], Lianzhang Wang[1] & Yi Zhong[1]*

It is believed that long-term memory (LTM) cannot be formed immediately because it must go through a protein synthesis-dependent consolidation process. However, the current study uses *Drosophila* aversive olfactory conditioning to show that such processes are dispensable for context-dependent LTM (cLTM). Single-trial conditioning yields cLTM that is formed immediately in a protein-synthesis independent manner and is sustained over 14 days without decay. Unlike retrieval of traditional LTM, which requires only the conditioned odour and is mediated by mushroom-body neurons, cLTM recall requires both the conditioned odour and reinstatement of the training-environmental context. It is mediated through lateral-horn neurons that connect to multiple sensory brain regions. The cLTM cannot be retrieved if synaptic transmission from any one of these centres is blocked, with effects similar to those of altered encoding context during retrieval. The present study provides strong evidence that long-term memory can be formed easily without the need for consolidation.

[1] Tsinghua-Peking Center for Life Sciences, IDG/McGovern Institute for Brain Research and School of Life Sciences, Tsinghua University, 100084 Beijing, China. [2]These authors contributed equally: Bohan Zhao, Jiameng Sun. *email: zhongyi@tsinghua.edu.cn

Common sense believes that long-term memory (LTM) is difficult to form for it requires repeated efforts for acquiring. The consolidation theory suggests that LTM needs hours to convert labile memory to LTM[1–3]. This process requires the synthesis of new proteins that supports long-lasting changes in synaptic morphology[3–5]. In the case of aversive olfactory conditioning in *Drosophila*, the formation of LTM that lasts for at least 7 days requires spaced repetitive training that induces transcription factor-mediated multi-hour consolidation[3,5]. In contrast, the memory produced by single-trial training, yields only short-term, mid-term, and anaesthesia-resistant memory components that largely decay away within 24 h[3,6].

However, in the present work, we showed that the same single-trial training induces a latent memory component that lasts for at least 14 days without decaying. Such component was formed immediately without the requirement of consolidation time or new protein synthesis. To reveal this novel component, we had to reinstate the copper grid, which was used in the training tube for conducting electric shocks, into the testing tube. This makes the memory-encoding context be fully reinstated for recall. Reinstatement of the copper grid has been overlooked in all previous extensive publications concerning aversive classic olfactory conditioning in *Drosophila*[7–10]. Thus, what previous reported LTM, as well as short-term, mid-term, and anaesthesia-resistant components, represent context-independent memories that can be retrieved using the conditioned odour alone. This newly identified novel component is known as context-dependent LTM (cLTM) because its retrieval requires the full reinstatement of the training environment as well as the conditioned odour.

The first experimental demonstration of encoding-context effects on memory recall was a study that showed better recall of a word list when the subject was in the same environment (either on land or underwater) during both training and testing[11,12]. Similar effects were also found with other environmental cues, such as background music[13,14], background odour[15,16] and background colour[17,18] of surrounding context. A number of psychological theories have been proposed to explain how context facilitates memory retrieval[12,17,19–22]. Similar context-dependent memory is also observed in rodents[21–24]. These investigations reveal that the hippocampus plays a role in mediating context cues-facilitated retrieval.

Extensive molecular and anatomical characterization of aversive olfactory conditioning in *Drosophila* provides a unique opportunity for gaining such insights. Odour information is first relayed to projection neurons at the antennal lobe (AL) via sensory inputs. It then bifurcates into two higher brain centres: the mushroom body (MB) and the lateral horn (LH)[25–28]. To date, formation and retrieval of all reported olfactory memory components, which are context independent, have involved MB neurons, with short-term memory being processed through the γ lobe and LTM through α/β lobe[7,29,30] while LH neurons are believed mainly to process innate information, such as innate avoidance and attraction[25,31]. Such extensive understanding of the neural anatomy of olfactory memory, and of accessible genetic tools, allows to examine more closely the circuits that mediate cLTM retrieval.

## Results

### Reinstating the copper grid to the testing tube reveals cLTM.
All previous publications involving aversive olfactory conditioning in *Drosophila* have used behavioural-assay apparatuses and procedures derived from the same design principle[10]. That is, the flies are trained in a training tube with a copper grid surface that delivers electric shocks (Fig. 1a, left panel) and tested in testing tubes without the copper grid (Fig. 1a, middle panel). Thus, in no previous studies has the retrieval of aversive memory components required the presence of the copper grid. When the copper grid was reinstated to the testing tubes (Fig. 1a, right panel)—a procedure that does not affect odour acuity (Supplementary Tab. 1) or lead to false memory performance (Supplementary Fig. 1a), the behavioural assay revealed striking effects. Single-trial conditioning produced a copper grid-dependent memory component that sustained longer than those seen previously, even in spaced, repeated trials. Specifically, the memory lasted for at least 14 days, which was the longest period tested (Fig. 1b).

To determine whether this copper grid-dependent memory enhancement reflects the general effects of context reinstatement, we tried to alter other elements of the training environment, specifically the colour of the surrounding light and the environmental temperature as flies are capable of responding to both[32,33]. When the red training light was switched to yellow in testing, or vice versa, memory enhancement failed to occur, even when the copper grid was provided (Fig. 1c and Supplementary Fig. 1c). Similarly, enhancement disappeared when the testing temperature was markedly different from the learning temperature (23 °C vs. 32 °C, or vice versa; Fig. 1c and Supplementary Fig. 1d). Therefore, changing any environmental condition of the encoding context completely blocked the copper grid-dependent memory.

However, the difference between the encoding environment and the testing environment had to be sufficiently significant or easily detectable to affect retrieval of the copper grid-dependent memory. For instance, memory enhancement was preserved when the testing temperature was changed from an encoding temperature of 23 °C to a testing temperature of 25 °C (Supplementary Fig. 1e).

In any case, retrieval of copper grid-dependent memory requires conditioned odour and full reinstatement of the encoding environmental context. This is why we termed this memory component as context-dependent LTM (cLTM).

### cLTM requires no protein synthesis-dependent consolidation.
Since most studies have only elicited LTM using spaced training protocols, and because LTM depends on protein synthesis[3], we further determined whether protein synthesis-dependent consolidation is necessary for cLTM. Remarkably, formation of such long-lasting memory was independent of protein synthesis, because administration of cycloheximide (CXM), a protein synthesis inhibitor, had no impact on cLTM formation (Fig. 1d), whereas the same treatment blocked LTM formation (Supplementary Fig. 1f), as expected[3,8,9]. In support of this observation, inhibition of protein synthesis through pan-neuronal expression of RICIN[34], a protein inactivating eukaryotic ribosomes, in transgenic flies (UAS-*RICIN*;nSyb-Gal4) showed no effect on cLTM formation either (Supplementary Fig. 1g). We confirmed such independency further through pan-neuronal expression (UAS-*dCREB2b*;nSyb-Gal4) of a repressor isoform of cAMP-response element-binding protein 2 (CREB2b) that is reported to block LTM[5] in transgenic flies had no impact on cLTM (Fig. 1g). Moreover, classical learning and memory *rutabaga* (*rut*) mutants, *rut1* and *rut2080*, with attenuated cAMP synthesis performed normal cLTM (Fig. 1h). Thus, data presented strongly suggest that cLTM formation requires no protein synthesis and therefore is different from the context-independent LTM. cLTM is also distinguishable from anaesthesia-resistant memory (ARM) for it remains normal in a radish mutant (Supplementary Fig. 1h) while ARM is impaired[35].

To validate these surprising findings, we determined whether the formation of cLTM takes time, another indication of

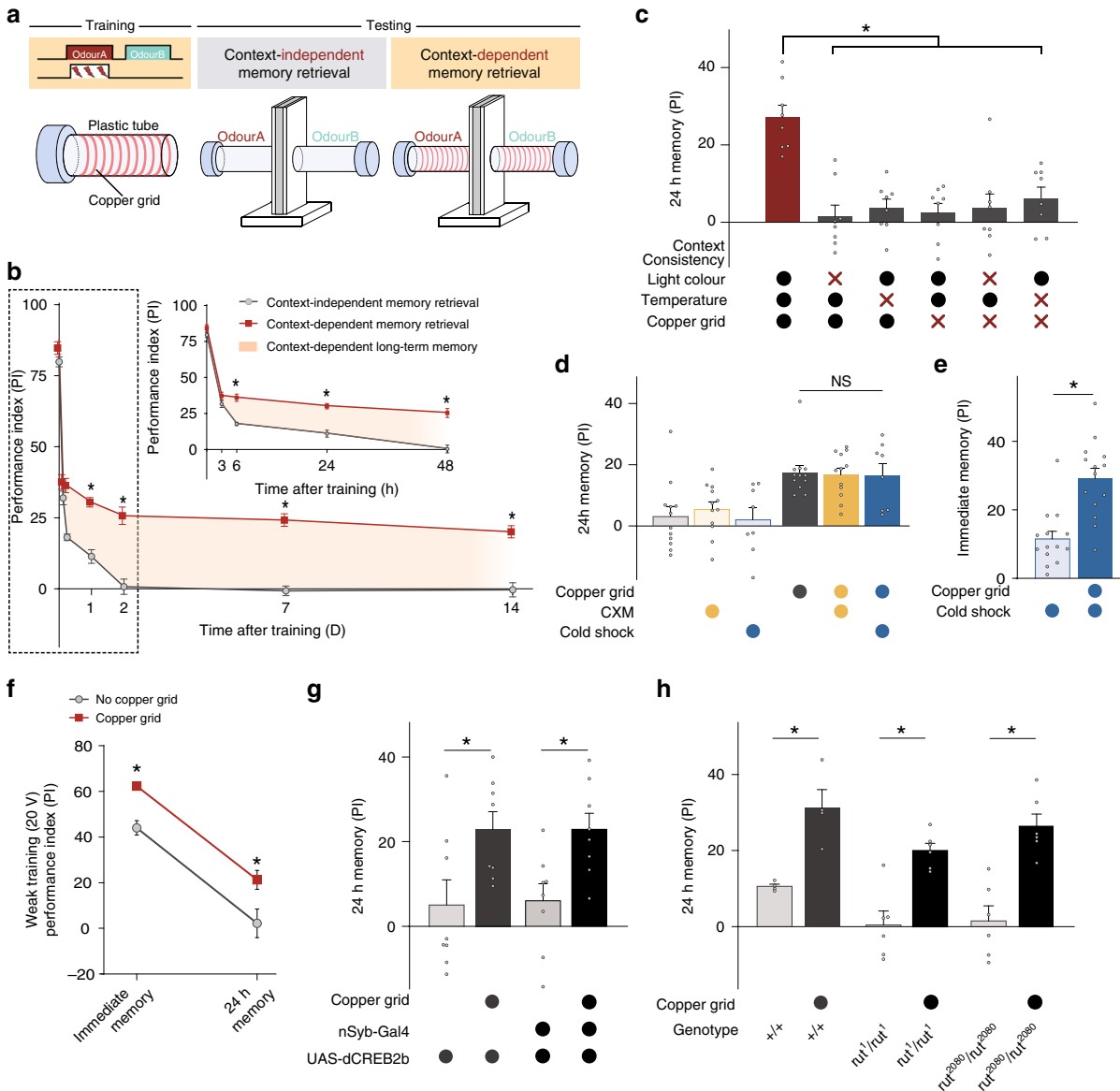

**Fig. 1** Reinstating encoding context makes cLTM retrievable. **a** Basic experimental schemes. Left: aversive olfactory classical conditioning. The training tube contains a copper grid surface to deliver the electric shock. Middle: classical memory testing in the T-maze. Right: modified testing with context reinstatement. The testing arms contain a copper grid to reinstate training context. **b** Memory retention curves tested with different methods. Context reinstatement enables context-dependent long-term memory (cLTM) to be retrieved using conditioned odours. The cLTM is measurable 3 h after training and lasts for at least 14 days without decay ($n = 10$–$12$). **c** cLTM retrieval requires multiple contextual conditions all matched. Any inconsistent contextual condition (i.e. light colour, temperature or absence of copper grid) abolishes cLTM retrieval ($n = 8$). Crosses denote different conditions from training and circles denote same conditions. **d** Protein synthesis inhibitor (cycloheximide) and cold-shock treatment cannot destroy cLTM ($n = 8$–$12$). **e** Copper grid improves 3-min memory performance after cold shock ($n = 14$). **f** Both immediate memory (3 min) and 24-h memory are significantly improved when the copper grid is present after weak training with a 20-V electric shock, which avoids the ceiling effect that occurs immediately after norma training ($n = 8$–$12$). **g** cLTM was not impaired in nSyb-Gal4;UAS-dCREB2b ($n = 8$). **h** cLTM was not impaired in $ru^1$ and $rut^{2080}$ mutants ($n = 4$–$6$). In all figures data show mean performance indices ± SEM; individual data points are displayed as dots. Asterisks denote a significant difference (*$P < 0.05$ by ANOVA or $t$ test)

consolidation. To this end, we characterized the resistance of cLTM to cold-shock treatment, which is known to abolish short-term and mid-term memory[3,8]. Twenty-four hours after training, cLTM remained unaffected (Fig. 1d) by typical cold-shock treatment. Such cold-shock resistance allowed us to perform two follow-up experiments:

Firstly, we applied cold-shock treatment for 2 min immediately after one conditioning trial. After 3 min of rest from the cold shock, a behavioural assay showed that the cold-shock-resistant cLTM was already formed in full strength (about 20%

of the performing index; Fig. 1e). Secondly, to further validate the observation, we reduced the strength of the training electric shock from 60 to 20 V to avoid any ceiling effects of memory strength. We found that, even at such a weak training strength, cLTM was formed immediately, because similar enhancements were immediately present in memory, indicating that cLTM formation that was sustained for a long time without decaying (Fig. 1f). Thus, cLTM is formed within 3 min after training, suggesting that no protein-synthesis consolidation is required for its formation.

This surprising observation prompted us to investigate whether cLTM is different from traditional LTM or merely constitutes the same memory retrieved in different environmental contexts. Multiple lines of evidence, presented below, suggested that cLTM is a distinct memory component with different molecular and anatomical features.

**Dopaminergic neurons are involved in cLTM formation**. LTM formation requires dopaminergic neurons (DANs), so we examined the role of DANs in cLTM encoding, comparing 24-h memory in control flies with that in flies whose synaptic outputs from DANs were blocked during training. For this purpose, expression of UAS-Shibire[ts1] (Shi[ts]) was targeted to DANs through TH-Gal4, so that normal synaptic output was allowed at permissive temperatures (23 °C) but blocked at restrictive temperatures (32 °C)[36]. To ensure consistent environmental conditions between training and testing, we adopted a strict regimen for temperature treatments. Specifically, to block synaptic transmission during training, flies were moved into a 32 °C environment 30 min before training and back to 23 °C immediately before training. They then completed training within 5 min and were tested 24 h later at 23 °C. Within the given time window (5 min), synaptic transmission of neurons expressing Shi[ts] remained blocked (Supplementary Fig. 2a). Similarly, in the case of

assays that required neuron blockade during testing, flies were moved to a 32 °C environment before testing but were trained and tested at 23 °C (Fig. 2a). The results showed that blocking the release of neurotransmitter from TH-Gal4-labelled neurons impaired cLTM formation, suggesting that DANs are required for cLTM encoding. This conclusion was further supported by the behavioural assay, which showed that no cLTM occurred in the *Drosophila* D1 dopamine receptor (dDA1)-mutant flies (*dDA1-dumb2*)[37] or in flies with pan-neuronal knockdown of *dDA1* (UAS-dDA1-RNAi;nSyb-Gal4) (Fig. 2b). This suggests that dDA1-mediated neuromodulation plays a role in cLTM acquisition.

However, dDA1s expressed in MB neurons were not involved in cLTM acquisition, because targeted overexpression of *dDA1* in MB neurons on a *dDA1dumb2* mutant background (*dDA1dumb2*; OK107-Gal4) failed to rescue cLTM acquisition. Consistent with this, *dDA1* knockdown in MB neurons (OK107-Gal4;UAS-dDA1-RNAi) did not affect cLTM. These data suggest that cLTM is encoded by dDA1-mediated neuromodulation, but not in the MB.

**Retrieval of cLTM is independent of mushroom-body neurons**. Interestingly, dDA1s in MB neurons were not involved in cLTM acquisition, while all previous studies in this field have

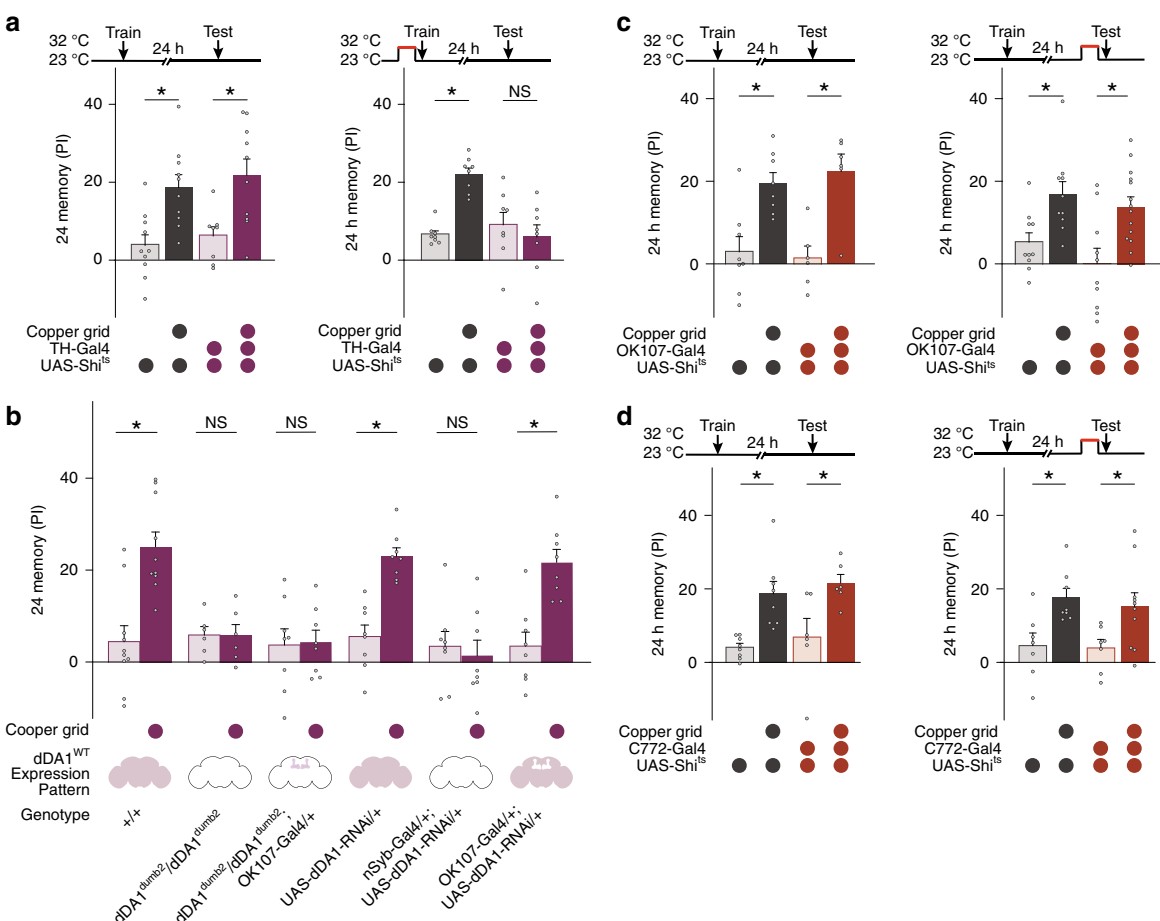

**Fig. 2** cLTM requires dopaminergic neurons, but not the mushroom body. **a** Top: protocols. Temperature shifting is finished immediately before training to avoid inconsistency in temperature conditions. Bottom: blockade of dopaminergic neurons with TH-Gal4 and UAS-Shits during training abolishes context-dependent long-term memory (cLTM) formation (n = 8–10). **b** Both mutant (*dDA1dumb2*) and knockdown dDA1 in pan neurons (nSyb-Gal4;UAS-dDA1-RNAi) abolish cLTM formation. Selective overexpression of dDA1WT in mushroom body (MB) in *dDA1dumb2* flies (dumb2;OK107-Gal4) does not rescue cLTM. Selective knockdown of dDA1 in the MB (OK107-Gal4;UAS-dDA1-RNAi) does not impair cLTM (n = 6–10). **c**, **d** Top: protocols. Temperature shifting is finished immediately before testing. Bottom: MB output is dispensable during cLTM retrieval (n = 6–14). Data are mean performance indices ± SEM; individual data points are displayed as dots; *P < 0.05 by ANOVA or *t* test

found that formation of context-independent, aversive memory components, including traditional LTM, involves MB neurons[38–41]. To confirm this observation, we examined the roles of MB neurons in cLTM retrieval. To this end, expression of UAS-*Shi*[ts] was targeted to MB neurons through two independent Gal4 drivers: OK107-Gal4 and C772-Gal4 (Supplementary Fig. 2b, c). Although LTM retrieval failed in OK107-Gal4;UAS-*Shi*[ts] flies (Supplementary Fig. 2d), cLTM remained intact in OK107-Gal4;UAS-*Shi*[ts] and C772-Gal4;UAS-*Shi*[ts] flies (Fig. 2c, d), confirming that MB neurons are not involved in the formation or retrieval of cLTM.

**Retrieving cLTM requires AL and projection neurons.** To identify which brain regions are required for cLTM retrieval, we investigated the role of AL local neurons and projection neurons (PNs). Olfactory information in flies is relayed from sensory neurons to AL neurons and PNs which then bifurcate to the MB and LH[42]. We first tested the effects of blocking the synaptic output from AL local neurons labelled by OK66-Gal4 (Supplementary Fig. 3a). Blockade of synaptic transmission at restrictive temperature abolished cLTM in OK66-Gal4; UAS-*Shi*[ts] flies (Fig. 3a). We then tested the effects of two distinct subgroups of projection neurons, with excitatory projection neurons (ePNs) projecting to both the MB and LH, labelled by GH146-Gal4 (Supplementary Fig. 3b), and inhibitory projection neurons (iPNs) projecting only to the LH region, labelled by MZ699-Gal4 (Supplementary Fig. 3c). The 24-h memory enhancement in the presence of grids was not evident when the output of either ePNs or iPNs was blocked (Fig. 3b, c). These observations demonstrate that AL neurons and PNs participate in olfactory information transmission during cLTM retrieval, as with all previously identified context-independent memory components. In contrast, iPNs labelled with MZ699-Gal4, which project to the LH, are required for olfactory habituation, but not retrieval of context-independent memory[28,43]. This effect of MZ699-Gal4 implies that the LH plays a role in cLTM retrieval.

**Retrieval of cLTM requires LH and AMMC neurons.** Interestingly, LH neurons are connected to multiple remote brain regions[25]. A recent discovery reports that the LH receives multisensory inputs from brain regions of various sensory systems[44]. These include the antennal mechanosensory and motor centre (AMMC), which communicates mechanosensory information, the ventral lateral protocerebrum (vlpr), which is responsible for colour vision[33], and other areas involved in taste and temperature[32,45]. We thus hypothesized that such converging neuronal connections mediate cLTM retrieval using multiple sensory modalities.

To test this hypothesis, we first focused on a subgroup of LH neurons linked to the AMMC. This centre receives diverse mechanosensory signals from Johnson's organ, including touch, hearing, proprioception, and wind sensing[46–49]. It then relays these signals to other brain regions, including the LH[25]. The expression pattern of NP1004-Gal4 was visualized by staining of the membrane target marker mCD8:GFP in NP1004-Gal4; UAS-*mCD8:GFP* flies (Fig. 4a, left panel). Indeed, AMMC-LH neurons labelled by NP1004-Gal4 and immunostaining showed that the presynaptic marker syt::GFP (a fusion of eGFP and the synaptic vesicle protein synaptotagmin) is enriched within the LH of NP1004-Gal4; UAS-*syt::GFP* flies (Fig. 4a, right panel), suggesting that there are synaptic connections from the AMMC to the LH.

Heat shock-induced reversible blockade of synaptic transmission impaired cLTM retrieval in NP1004-Gal4; UAS-*Shi*[ts] flies (Fig. 4b). Corroborating this observation, blockade of synaptic transmission in AMMC neurons labelled with R38E07-Gal4 and

NP0761-Gal4 also suppressed cLTM retrieval (Fig. 4c and Supplementary Fig. 4c). These results suggest that the representation of mechanosensory information within AMMC-LH neurons is critical for cLTM retrieval. To further confirm this conclusion, we blocked mechanosensory inputs via removing arista, which is a major mechanosensory organ in *Drosophila*, after training. Results showed that this treatment impaired cLTM (Fig. 4d) but had no impact on learning (Supplementary Fig. 4d), suggesting a role for AMMC-LH neurons in cLTM.

Next, we imaged calcium responses in the LH terminal of AMMC-LH neurons after mechanosensory stimulus applied with a tiny brush to the arista (see Methods section). To this end, we expressed GCamP6f, a calcium-sensitive fluorescent protein[50], driven by NP1004-Gal4. We then recorded the fluorescence of GCamP6f from the LH regions (Fig. 4e). There were robust responses to arista contact with the brush in the LH region, supporting the notion that mechanosensory information is relayed to the LH through AMMC-LH neurons.

To confirm that these observations were behaviourally significant, we monitored AMMC neuronal activity using the transcriptional reporter of intracellular calcium (TRIC)[51], which increases GFP expression in proportion to intracellular calcium levels in flies. The fluorescence of TRIC from the AMMC region was calculated 3 h after retrieval and normalized to control flies (Fig. 4f). Significantly greater TRIC signal was observed in the AMMC after context-dependent retrieval than in the control flies or after context-independent retrieval, showing that AMMC neuronal activity correlates well with context-dependent retrieval. Thus, LH neurons are capable of integrating mechanosensory information from the AMMC and olfactory information from the AL to retrieve cLTM (Fig. 4g).

**Multisensory integration in the LH underlies cLTM retrieval.** We then further verified whether other sensory systems were also involved in this process, such as the visual system. We blocked visual input through targeted expression of temperature-sensitive mutant *Shi*[ts] in eyes (UAS-*Shi*[ts]; GMR-Gal4) and the optic lobe neurons (UAS-*Shi*[ts];R82D10-Gal4) during cLTM retrieval (Supplementary Fig. 6a). cLTM were abolished in both cases, suggesting that the visual system is also involved in the cLTM retrieval.

Such visual input as well as other potential sensory inputs supposedly converges to LH neurons, just like in the case of mechanosensory input. We performed targeted expression of the presynaptic marker syt::GFP in available Gal4 lines, labelling the following LH neurons, superior medial protocerebrum to LH (smpr-LH; MZ671-Gal4), superior lateral protocerebrum (relevant to taste[52]) to LH (slpr-LH; NP3060-Gal4), and ventral lateral medial protocerebrum (relevant to visual[33]) to LH (vlpr-LH; NP5194-Gal4)[25]. The results showed that projections from targeted brain regions congregate, or make synapses, in the LH region (Fig. 5a), suggesting that diverse contextual information is relayed to the LH.

We then tested the effects of manipulating the labelled neurons on cLTM retrieval. Blocking synaptic transmission of each subgroup of LH neurons abolished cLTM retrieval (Fig. 5b and Supplementary Fig. 5). However, cLTM retrieval was not affected by blockade of MB output neurons (MB-V2, labelled by NP2492-Gal4) that project to the LH, which was reported required for traditional LTM[53]. This connection may be necessary for retrieval of context-independent aversive olfactory memory[53,54]. Thus, cLTM retrieval also involves integration of synaptic inputs from other distinct sensory brain regions to the LH (Fig. 5c).

To determine whether these LH neurons are also involved in retrieval of the traditional LTM, we blocked these neurons during

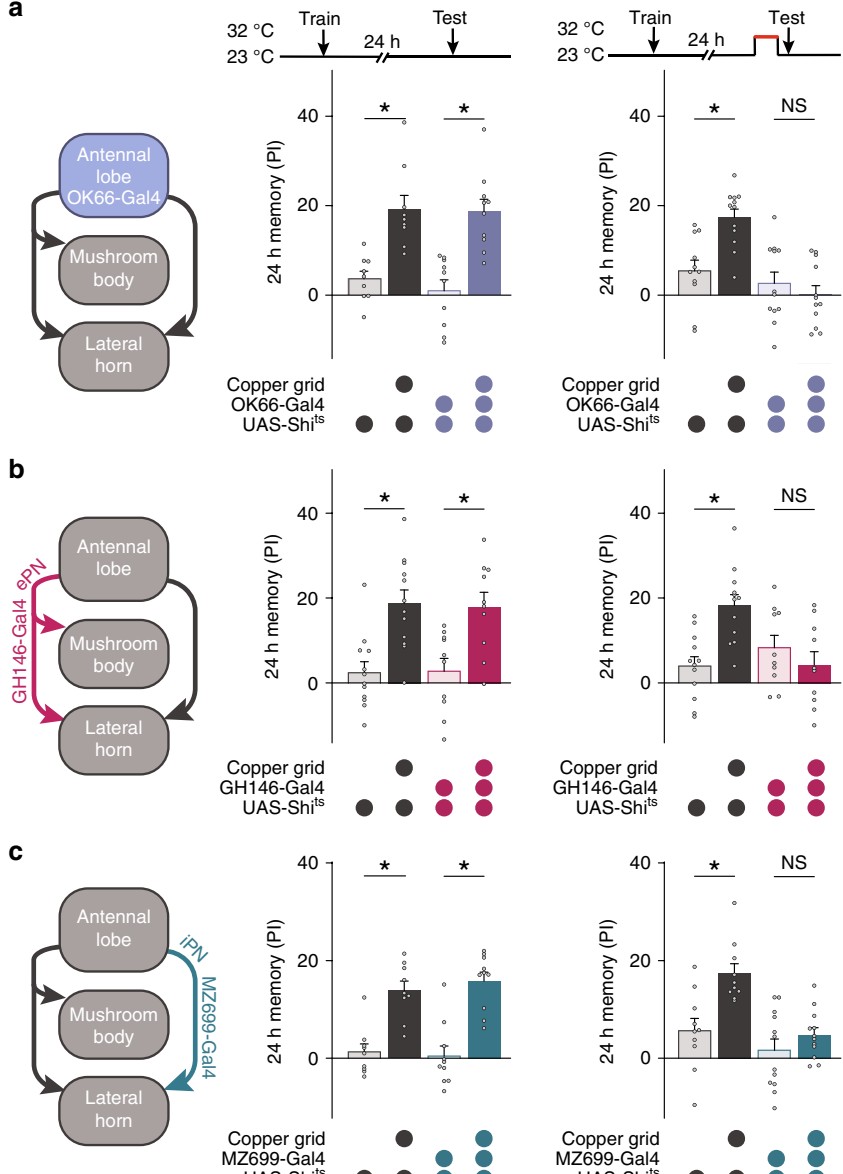

**Fig. 3** Retrieval of cLTM requires the antennal lobe and projection neurons. **a** Left: Schematic of OK66-Gal4 antennal lobe (AL)-local neurons. Right: Blockade of OK66 neurons during testing abolishes cLTM retrieval (n = 9–12). **b** Left: Schematic of GH146-Gal4 excitatory projection neurons (ePNs). Right: blockade of GH146 neurons during testing abolishes cLTM retrieval (n = 10–12). **c** Left: Schematic of MZ699-Gal4 inhibitory projection neurons (iPNs). Right: Blockade of MZ699 neurons during testing abolishes cLTM retrieval (n = 9–12). Data are mean performance indices ± SEM; individual data points are displayed as dots; *P < 0.05 by ANOVA or t test

retrieval after spaced training. Such blockade exerted no impact on LTM (Supplementary Fig. 5b).

To further validate a role of LH neurons in retrieval of cLTM, we then tested the effects of blocking LH output neurons. A number of Gal4 strains are identified to label LH output neurons[44]. The 24-h cLTM was not evident when the output from neurons labelled by PV5b3, AD1d1, AV4b4/c1, PV5g1/g2, or AV6b1 was blocked (Supplementary Fig. 5c), while AD1e1 and AV6a1 were not. These observations demonstrate that LH plays a central role in cLTM retrieval.

Taking the data presented together with the reported study of context-independent memory components, we are led to propose a model for cLTM and LTM retrieval (Fig. 6). Multisensory integration in the LH gates the retrievability of cLTM while conditioned odour alone is sufficient in the retrieval of context-independent memories.

## Discussion

In the present study, we identified and analysed the cLTM, which can be observed by reinstalling the copper grid in testing tubes (Fig. 1a, b) and thus fully reinstating the environmental context of training for memory retrieval (Fig. 1c). The most striking finding of the study was corroborative data suggesting that long-lasting memory is formed immediately after a single trial of training, without the need for protein synthesis-dependent consolidation. Three lines of evidence in support of this conclusion are outlined below.

Firstly, cLTM formation is independent of protein synthesis. cLTM formation is not impaired by blocking protein synthesis through either administration of a protein synthesis inhibitor (cycloheximide) (see Fig. 1d), pan-neuronal expression of RICIN (see supplementary Fig. 1g), or pan-neuronal expression of suppressor of transcription factor, CREB2b (see Fig. 1g). These same

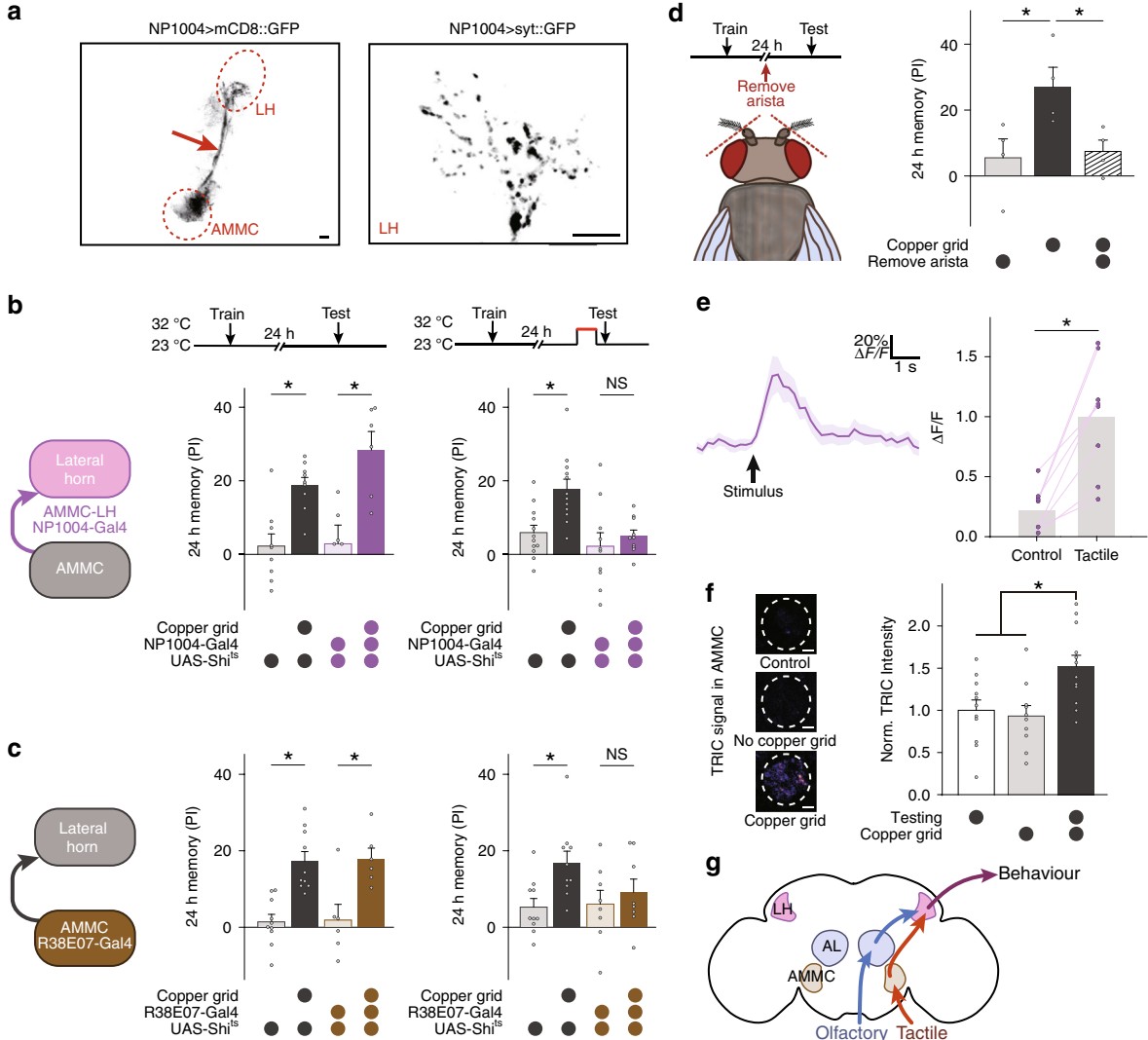

**Fig. 4** Retrieval of cLTM requires AMMC and AMMC-LH neurons. **a** Left: The expression pattern of NP1004-Gal4. The neurons that connect the antennal mechanosensory and motor centre (AMMC) and lateral horn (LH) are broadly labelled (arrow). Right: The presynaptic marker syt::GFP driven by NP1004-Gal4 is highly concentrated in the LH region. Scale bar = 20 μm. **b** Left: Schematic of R38E07-Gal4 AMMC neurons. Right: Blockade of R38E07 neurons during testing abolishes cLTM retrieval (*n* = 6–12). **c** Left: Schematic of NP1004-Gal4 AMMC to lateral horn neurons. Right: Blockade of NP1004 neurons during testing abolishes cLTM retrieval (*n* = 6–10). **d** Left: Protocol and experimental setup of arista lesion. Right: Removal of the arista abolishes cLTM retrieval (*n* = 4). **e** In vivo calcium imaging shows that GCaMP6f fluorescence, driven by NP1004-Gal4, induces calcium responses to tactile stimulation of the arista in the lateral horn (LH) region (*n* = 8). Left: Time course averaged across all animals. The arrow indicates delivery of the tactile stimulus. Right: The integrated peaks of ΔF/F in the time bins. The responses to tactile stimulus were significantly higher than the control group. **f** Left: Samples of transcriptional reporter of intracellular calcium (TRIC)-labelled R38E07 neurons in AMMC after different treatments: direct measurement, test without copper grid, and test with copper grid 24 h after training. Scale bar = 20 μm. Right: Normalized TRIC intensity calculated with different treatments (*n* = 10–12). **g** Schematic of the brain showing a model of cLTM retrieval mediated by olfactory and tactile information. Data are mean results ± SEM; individual data points are displayed as dots; *P < 0.05 by ANOVA or *t* test

treatments are reported to abolish the formation of traditional LTM[3,34,55], which is induced through repetitive training[3]. Although single trial training is capable of inducing traditional LTM, such as in the tasks of hunger stress-facilitated aversive olfactory memory[56] and appetitive olfactory memory[57] in *Drosophila*, as well as contextual memory in rodents[55], all these LTM components so far reported are protein synthesis-dependent and are abolished by protein synthesis inhibitor.

Secondly, cLTM takes almost no time to form. We presented two independent lines of evidence for this: (1) cold-shock treatment, which removes immediate memory, revealed that cLTM is formed immediately after training (Fig. 1e), or at least within 5 min. Specifically, cold shock was given for 2 min immediately

after training, and testing was performed after 3 min rest upon completion of cold shock; (2) with reduced training strength, which prevents ceiling effects, cLTM was shown to occur immediately after training, without any decay (Fig. 1f). In previous work, the same cold-shock treatment has shown that both aversive and appetitive LTMs take hours to consolidate[3,57].

Thirdly, cLTM is a novel memory component distinct from traditional LTM, because it is formed, stored, and retrieved from different neural networks. Although both require activation of dopamine receptor-mediated signalling pathways to form (Fig. 2a), LTM involves MB neurons[58,59], while cLTM does not (Fig. 2b). Moreover, LTM retrieval is mediated by MB neurons[8,30], while cLTM requires no MB (Fig. 2c, d), but is

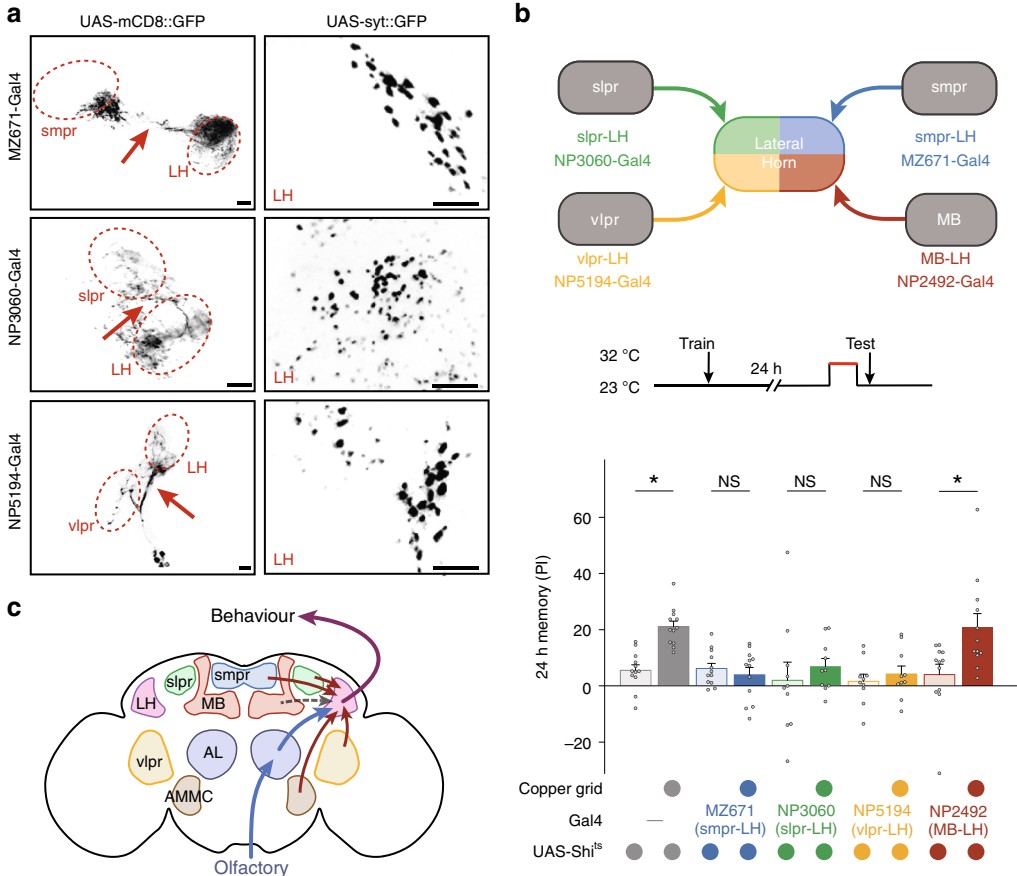

**Fig. 5** Retrieval of cLTM requires neurons connecting the LH with other regions. **a** Left: NP1004-Gal4, MZ671-Gal4, NP3060-Gal4, and NP5194-Gal4. Neurons connecting the superior medial protocerebrum, superior lateral protocerebrum, and ventral lateral protocerebrum with the LH are broadly labelled (arrows). Right: The presynaptic marker syt::GFP in these LH neurons is highly concentrated in the LH region. Scale bar = 20 μm. **b** Top: protocol. Middle: Schematics of MZ671-Gal4, NP3060-Gal4, NP5194-Gal4, and NP2492-Gal4, that labels neurons connecting the smpr, slpr, vlpr, and mushroom body (MB) with the LH. Bottom: Blockade of any of these LH neurons during retrieval abolishes context-dependent long-term memory (cLTM), but MB-V2 (connecting the MB to the LH) is dispensable during cLTM retrieval (n = 10–12). **c** Schematic of the brain showing a model of cLTM retrieval mediated by multiple information integration. Grey-dashed arrows denote that information from the MB is not required. Data are mean performance indices ± SEM; individual data points are displayed as dots; *P < 0.05 by ANOVA or t test

mediated by LH neurons that connect with multiple remote brain regions implicated in the perception of environmental conditions (Figs. 3–5)[25]. Significance of these findings is further discussed below.

Retrieval of cLTM requires full reinstatement of the encoding environmental context, including multiple sensory cues. How do multiple sensory modalities providing information about environmental context together support cLTM retrieval? Based on the model of context-dependent and context-independent memories we proposed (Fig. 6), reinstatement of all context factors is necessary for cLTM retrieval, which requires various sensory centres to acquire multiple modalities of environmental context. LTM retrieval requires only task-relevant cues or a conditioning stimulus. In the present case, odour was sufficient to retrieve LTM from MB neurons (Fig. 6, top panel). However, projections from the AL and multiple other brain sensory centres to the LH work together in the retrieval of cLTM, similar to the AND gate in logic (Fig. 6, bottom panel), which means any missing input leads to the failure. Like the AMMC-LH pathway, these sensory signals are then relayed through LH neurons and finally integrated in the LH region to facilitate cLTM retrieval. Thus, in the reinstated context, the conditioning stimulus is able to retrieve cLTM. However, in the case of LTM retrieval, only the conditioning stimulus is sufficient.

In psychology, the term context effect refers to the phenomenon whereby memory is better retrieved in the encoding context[11]. A number of hypotheses have been advanced to explain this observation[11,17,20,60]. The most widely accepted one is the Specific Encoding Principle[19], which claims that context is encoded as a cue within the memory traces. As the memory gradually becomes unclear, it comes to be context-dependent. This process is referred as cue-dependent forgetting[22,61]. However, the present study suggested that after an event, two types of associative memories are encoded; one is context-independent and the other is context-dependent. Context-dependent memory is not transformed from forgotten memory as psychological hypotheses suggests, but is formed immediately after training. Thus, the enhanced memory performance observed in the reinstated context is actually cLTM.

The protein synthesis-dependent consolidation has long been seen as the foundation for formation of long-lasting memories[3–5,62], because it is necessary to support stable morphological changes of synapses. The finding that cLTM is formed immediately without any protein synthesis-dependent consolidation process imposes a challenge to memory consolidation theory. In fact, a number of reports suggest that the protein synthesis seems not required for LTM storage[63–68]. Although the current study shows neural modulation activated through

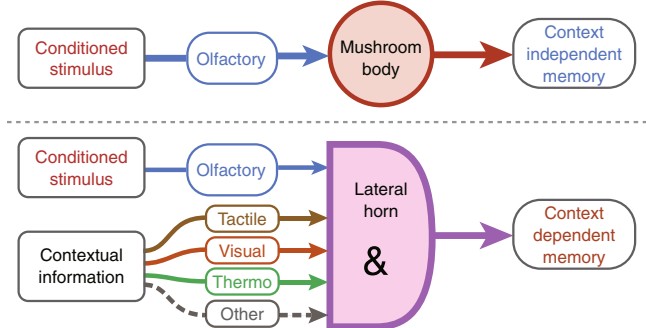

**Fig. 6** Model of cLTM retrieval: Multisensory integration in the LH. Top: Conditioning stimulus (olfactory cue) is sufficient to retrieve context-independent memories in the MB. Bottom: Both conditioning stimulus and contextual information are transported to the LH, mediating cLTM retrieval together. Only when all modalities of contextual information are integrated in the LH and matched to the encoding context can the cLTM be retrieved using the conditioning stimulus, much like the AND gate in logic

dopamine receptors is required for cLTM formation, the nature of such modulation remains to be determined. Protein synthesis-independent modulation through phosphorylation is capable of mediating syntactic plasticity and memory formation. However, such mechanism can only last for a few hours[69]. It would be of great interest to study the molecular and cellular mechanisms that support formation of long-lasting memory in a protein synthesis-independent manner.

## Methods

**Fly strains**. All flies (*Drosophila melanogaster*) were raised on standard cornmeal medium at 23 °C and 60% relative humidity under a 12-h light−dark cycle. The control strain was *w1118(isoCJ1)*. *UAS-Shibire[ts1]*, *OK66-Gal4*, *GMR-Gal4*, *rut1*, *rut2080*, *radish*, *dumb2*, and *MB-Gal80* were extant stocks in the laboratory. The following *Gal4* lines were obtained from the Department of *Drosophila* Genomics and Genetic Resources of the Kyoto Institute of Technology (DGGR-Kyoto): *NP1004-Gal4* (stock #112440), *NP3060-Gal4* (#104359), *NP5194-Gal4* (#113625) and *NP2492-Gal4* (#104219). *R38E07-Gal4* (stock #50007), *nSyb-Gal4* (#51635), *R82D10-Gal4* (#40145), *R44G08-Gal4* (#50216), *R28A10-Gal4* (#48074), *R55C09-Gal4* (#39107), *R54G12-Gal4* (#41280), *R53B06-Gal4* (#38863), *R94B04-Gal4* (#40675),

*UAS-GCamP6f* (#42747), *UAS-syt::GFP* (#6925), and *UAS-p65aD::CaM;UAS-MKII::nlsLexADBDo;LexAop2-mCD8::GFP;UAS-mCD8::RFP* (#62827) were obtained from the Bloomington *Drosophila* Stock Center (BDSC; Indiana University, Bloomington, IN, USA). *UAS-dDA1-RNAi* (TH04278.N) was acquired from Tsinghua Fly Center (Tsinghua University, Beijing, China). *GH146-Gal4* and *MZ671-Gal4* were gifts from Yulong Li, Peking University; *UAS-RICIN* was a gift from Ann-Shyn Chiang, National Tsing Hua University; *NP0761-Gal4* was a gift from Wei Zhang, Tsinghua University; *MZ699-Gal4* was a gift from Li Liu, Institute of Biophysics, Chinese Academy of Sciences; *UAS-dCREB2b* was a gift from Ronald Davis, Scripps Research Institute. For experiments with *NP1004-Gal4*, and *NP2494-Gal4*, which are inserted on the X chromosome, only females of *Gal4/+; UAS-Shits/+* were used to calculate the memory index. The following strains used for the memory experiments were outcrossed to the w1118 background: *TH-Gal4*, *OK107-Gal4*, *C772-Gal4*, *nSyb-Gal4*, *OK66-Gal4*, *GH146-Gal4*, *MZ699-Gal4*, *R38E07-Gal4*, *GMR-Gal4*, *UAS-Shibirets*, *UAS-dDA1-RNAi*. However, the following strains used for the memory experiments were not outcrossed, including *NP1004-Gal4*, *MZ671-Gal4*, *NP3060-Gal4*, *NP5194-Gal4*, *NP2492-Gal4*, because all these strains are used for comparison only in the context of the same genotype but at different conditions.

**Behavioural assays**. For behavioural experiments, males from *Gal4* lines were crossed to *UAS-Shits* females. For heterozygous controls, *UAS-Shits* flies were crossed to *w1118*. All flies were raised at 23 °C and mixed sex populations of 2–5-day-old flies were used in all experiments. The Pavlovian olfactory aversive conditioning procedure was performed in a behavioural room at 23 °C and 60% relative humidity.

During training, roughly 80 flies successfully received the following stimuli in a training tube, which contained a copper grid: air for 90 s, an odour paired with 12 pulses of 60 V electric shock (CS+) for 1 min, air for 45 s, a second odour without pairing the electric shock (CS−) for 60 s, and finally air for 45 s. 3-octanol (OCT, 15 μl in 10 ml mineral oil; Sigma-Aldrich, St. Louis, MO, USA) and 4-methylcyclohexanol (MCH, 10 μl in 10 ml mineral oil; Sigma-Aldrich) were used as

standard odourants. This process describes a typical training session. Four sequential cycles with 15-min intervals constitute spaced training.

For context-dependent memory retrieval, trained flies were transferred into a T-maze, where they were allowed 1 min to choose between two odours (CS+ and CS−). Meanwhile, the testing arms contained the copper grid to reconstruct the context of the training tube. Flies were exposed to empty arms during context-independent memory retrieval.

Memory retention was quantified by a PI calculated based on the fraction of flies in the two T-maze arms. A PI of 100 indicated that all flies made the right choice to avoid the odour paired with the electric shock, while a PI of 0 indicated no memory retention, as reflected by a 50:50 distributions between the arms. To balance naïve odour bias, two reciprocal groups were trained and tested simultaneously. One group was trained to associate OCT with an electric shock, and the other was trained to associate MCH with an electric shock. The complete PI was defined as the average PI of the two groups. For 3-min memory, flies were tested immediately after training. For longer memory retention, flies were placed in a fresh vial with the same contents as the vial they had been kept in before the training until the test.

For cold-shock anaesthesia, flies were transferred into pre-chilled plastic vials and kept on ice for 2 min.

For cycloheximide (CXM) feeding, flies were provided food with (CXM+) or without (CXM−) 35 mM CXM (Sigma-Aldrich) dissolved in control solution, 5% (wt vol$^{-1}$) glucose, and 3% (vol vol$^{-1}$) ethanol, for 2 days before and after training until memory retention was tested.

For neural inactivation experiments using *UAS-Shits*, crosses were reared at 23 °C to avoid unintended inactivation. Because the inhibition induced by *Shits* lasts for over 5 min, flies were shifted to 32 °C for 30 min immediately before behavioural training/tests, to avoid changing the context temperature.

For protein synthesis-inhibition experiments using *UAS-RICIN*, crosses were reared at 18 °C to avoid unintended inhibition. Flies were shifted to 30 °C immediately after training until test.

For context dependency experiments with coloured light, an LED spot with a 3.0 V button battery was fixed in the air inlet side of the training or testing tube. Red light was delivered by an LED spot with 625 nm wavelength. Yellow light is delivered by an LED spot with 590 nm wavelength.

For context dependency experiments with temperature, flies were first transferred to a behavioural room at 23 °C or 32 °C and 60% relative humidity and allowed to adapt to the environment for 15 min.

**Immunohistochemistry**. Flies were quickly anesthetized on ice and whole brains were dissected in ice-cold PBS within 5 min, then stained as described. Brains were fixed in 4% paraformaldehyde in PBS for 30 min on ice. Brains were incubated for at least 72 h with the primary antibodies anti-GFP (chicken, 1:2000; Abcam, Cambridge, UK), anti-nc82 (mouse, 1:10; Developmental Studies Hybridoma Bank, Iowa City, IA, USA) and anti-DsRed (rabbit, 1:500; Takara Bio, Kyoto, JP). Brains were washed three times again in PBS with 0.2% Triton X-100 and transferred into secondary antibody solution (anti-chicken Alexa Fluor 488, 1:200; anti-mouse Alexa Fluor 647, 1:200; anti-rabbit Alexa Fluor 647, 1:200; Molecular Probes, Eugene, OR, USA) and incubated for 48 h at 4 °C. Images were obtained using a Zeiss LSM710 confocal microscope (Carl Zeiss AG, Oberkochen, Germany).

**In vivo two-photon calcium imaging**. Three- to seven-day-old female flies were used. After anaesthetized in a plastic vial on ice for 15–20 s, flies were then gently inserted into a hole of a thin plastic rectangular plate. Fly was stabilized in the hole by glue, and its legs were stuck to avoid touching the arista. In a saline bath, the area surrounding the region of interest was surgically removed to expose the dorsal side of the brain. Fat and air sacs were gently removed to give a clear view of the brain. A tiny brush was placed directly under the arista of the fly. The tiny brush can be changed up and down to allow in to touch the arista. For calcium- response imaging, the ×40 water immersion objective lens (NA = 1.0; Zeiss) was lowered near the exposed brain, while the underside of the plastic specimen mount was kept dry and the arista can be touched by the tiny brush in a more natural situation.

Imaging was performed on a Zeiss LSM 7MP two-photon laser scanning microscope with an imaging wavelength at 910 nm (Carl Zeiss AG). The 512 × 512 pixel images were acquired at 2.6 Hz. In each trial, 50 s of baseline was recorded, followed by a brush stimulus recording. Two to three minutes of rest were given in between trials when multiple trials were applied. GCaMP responses and the peak value within 2 s before and after stimulus were quantified using custom software written in MatLab (MathWorks, MA, USA). For the brain region of interest during the experimental period, the average fluorescence value, $F_{av}$, was then converted to $\Delta F/F$ using the formula $\Delta F/F = (F_{av} - F_0) F_0^{-1}$, where $F_0$ is the baseline fluorescence value, measured as the average of 0–40 s.

**TRIC analysis**. For TRIC signal analysis, *R58E07-Gal4* flies (BDSC; #50007, see above) were crossed with a TRIC line (#62827) to quantify the changes of intracellular $Ca^{2+}$ levels in AMMC during CIM and CDLM retrieval. Brain tissues from 2 to 5-day-old flies were processed for anti-GFP similarly as described above. Brains were fixed in 4% paraformaldehyde in PBS for 30 min on ice. Brains were incubated for at least 24 h with the primary antibodies anti-GFP and anti-DsRed.

Brains were washed three times again in PBS with 0.5% Triton X-100 and transferred into secondary antibody solution and incubated overnight at 4 °C. Images were obtained using a Zeiss LSM710 confocal microscope with ×40 water immersion objective lens (NA = 1.0; Zeiss). The region of interest (ROI) was localized with DsRed signal. To quantify GFP fluorescence, the sums of all pixels of consecutive stacks (1.0 $\mu$m-thick each) comprising the ROI were calculated. Background intensity adjacent to the ROI was measured and subtracted. The fluorescence intensity, areas and volumes of ROI were measured using Imaris (version 9.2; Bitplane, Zurich, CH). Then the relative TRIC intensity was calculated by normalizing the control group as 1.0.

**Statistics**. Statistics were performed with GraphPad Prism software (version 7; GraphPad Software, San Diego, CA, USA). All data satisfied the assumption of normal distribution (one-sample Kolmogorov–Smirnov test). Comparisons between two groups were performed using two-tailed $t$ tests. Comparisons of multiple groups were performed using one-way or two-way analysis of variance (ANOVA) tests followed by Bonferroni correction for multiple comparisons. $P$ values less than 0.05 were considered statistically significant and are marked with an asterisk in figures; NS indicates nonsignificant differences ($P > 0.05$).

**Reporting summary**. Further information on research design is available in the Nature Research Reporting Summary linked to this article.

## Data availability
The data supporting the findings of this study are available from the corresponding author upon reasonable request.

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

## Acknowledgements
We thank DGGR-Kyoto, BDSC, Tsinghua Fly Center, Y. Li, W. Zhang, Y. Rao, L. Liu, Y. Zhu, and R. Davis for fly stocks; Y. Zhang, B. Lei, Y. Chen, W. Hu, L. Zhang for discussions; and all members of the Zhong lab for their support. This work was supported by grants from the National Science Foundation of China (91332207 and 91632301, to Y.Z.), the Beijing Municipal Science and Technology Commission (Z161100002616010, to Y.Z.), the National Science Foundation of China (31700912, to Q.L.) and the China Postdoctoral Science Foundation (2016M601004, to Q.L.).

## Author contributions
B.Z., X.Z., J.S. and Y.Z. conceived and designed this project. B.Z. performed all behaviour experiments. J.S. guided imaging, performed live imaging experiments. B.Z., X.Z., J.S. and H.M. performed and characterized expression pattern. H.M. performed and analysed the TRIC signal. L.W. provided facilities and support for behavioural experiments. B.Z., J.S., Y.N. and Q.L. prepared fly stock. Q.L. provided intellectual contributions. The manuscript was written by B.Z., X.Z. and Y.Z.

## Competing interests
The authors declare no competing interests.
