## [Peer Review File · Nature Communications]

Reviewers' Comments:

Reviewer #1:

Remarks to the Author:

This is a very interesting short manuscript from Zhao et al that shows the clear importance for context in the formation of LTM. The experiments are solid and clearly show that flies form a context dependent LTM (cLTM) after a single training trial. I think this alone is worthy of publication.

Surprisingly the authors show that this cLTM cannot be disrupted by cycloheximide feeding, or cold-shock applied immediately after training. These data are certainly suggestive that cLTM is quite unique and so they are important. They are however, rather traditional crude and don't allude to any detailed mechanism.

The authors do a few experiments to investigate the underlying neural mechanisms. They show that cLTM requires TH-Gal4 labelled neurons during training. This is a good start but obviously TH-GAL4 labels many neurons that have previously been implicated in aversive learning. This literature however focuses on the mushroom body (MB). Zhao et al contend that cLTM is independent of the MB because the DA receptor is not required in MB neurons and they can block MB output during retrieval, without consequence. Again these are really interesting results that imply that DA signals to elsewhere are critical. They do not investigate this further though.

Instead the authors follow up by showing that retrieval requires olfactory projection neurons. This seems obvious because it's hard to imagine how the odors would otherwise be recognized and they are presumably still the instructive cue even for cLTM. Other experiments show that some neurons in the lateral horn are also required to express cLTM. Again, these are interesting, although they seem rather preliminary.

So in summary, I find this an interesting short story with some good experiments. The results are worth publishing in their own right, but the current understanding of mechanism is rudimentary.

Reviewer #2:

Remarks to the Author:

In this manuscript, Zhao et al introduce a modification of the commonly used olfactory memory in flies by including a copper grid in the testing portion, thereby adding a context to the classically conditioned memory. Remarkably, the authors find that this modification not only improves the memory, but is not dependent on canonical memory machinery, primarily the mushroom body and independence from protein synthesis. These findings suggest plasticity/redundancy in olfactory memory circuits that has not been previously appreciated (though has been indicated by other results such as Dissel et al, 2015)). The findings have potential to broadly impact the field, and the way memory circuits are interpreted. That said, given the implications of this work, I believe there is added burden of proof and description of the phenotype that are required to fully establish the contextual long-term memory assay.

Show LH but where does memory come from in visual pathway? Where is the integration?

1. The introduction moves from flies to humans and back to flies, and could be refocused to setup the significance of the results section. Also, there is discussion of human contextual memory, but extensive work in rodents is not mentioned. It would be very helpful to add discussion of this alongside work in humans.

2. In figure 1C it would be useful to provide competing contexts. Trained odor with copper-grid vs. untrained odor with copper grid.
3. Were flies outcrossed into the control w1118 strain to control for background?
4. I do not like the use of 'Life-long memory.' Indeed 15 days is a long time for the fly, but it is not lifelong and it would be better to say something like protein synthesis- independent long-term memory.
5. The characterization of the sensory processes involved seems insufficient and jumps to central brain circuitry without a detailed characterization of sensory inputs. Were training and testing performed in the light or dark? What happens to vision mutants? mechanosensory mutants? The model in Figure 6 describes different inputs but these are not directly tested.
6. While the paper clearly shows cLTM is different from standard LTM, it would seem important to test canonical cAMP signaling pathways to see if these are shared.
7. My understanding is LTM is formed in a single trial using olfactory taste memory and mammalian shock memory. This may be important to reference.

Minor comments

1. The opening sentence (Line 25) seems subjective. It may be better to biologically define 'difficult to form'
2. Line 30, include 'spaced' ?
3. Line 64: 'innate' instead of 'memory-irrelevant'?
4. Line 367. Please describe the control solution for CXM.

Reviewer #3:

Remarks to the Author:

Zhao et al NCOMMS-19-03732-T

In this manuscript, Zhao et al propose formation of context-dependent LTM (cLTM) by single cycle training. They suggest the following significance of cLTM. 1) cLTM lasts more than 14 days after single-cycle training, 2) cLTM is protein synthesis independent, 3) mushroom body (MB) is dispensable for cLTM retrieval (also formation?), and 4) lateral horn (LH) is a center for cLTM retrieval (also formation?). While these findings are striking, there are some weaknesses and concerns that have to be addressed before publication.

Major concerns

- 1) Another protein synthesis inhibition and dose dependency of cycloheximide for cLTM formation: Authors only test cycloheximide without checking its dose dependency. Since this is a major claim of this paper, reviewer wants to know whether other protein synthesis inhibitors still do not affect cLTM formation. For example, RICIN employed by Chen et al Science (2012), is a gene encoding inhibitor of protein synthesis.
- 2) LH>shi(ts) experiments; Since authors claim that LH is a center of cLTM, reviewer needs to know whether inhibiting outputs from LH affect cLTM retrieval. Also, if LH is exclusively involved in cLTM (as shown in Figure 6), LH>shi(ts) should have no effects on classical LTM.
- 3) cLTM is a LTM?: Given that LTM is defined that it is depending on gene expression and protein synthesis, cLTM may be another form of anesthesia-resistant memory (ARM). Hence, it is worth to check cLTM in ARM mutants, such as radish.

Minor concerns

- 1) Reviewer curious to know if cLTM is also independent of transcription.
- 2) Among various contexts including vision, tactile, temperature, why only odor can form context independent memory?
- 3) Which brain regions require D1 function for cLTM formation?
- 4) Further mechanical insights into cLTM?

Reviewer #1:

This is a very interesting short manuscript from Zhao et al that shows the clear importance for context in the formation of LTM. The experiments are solid and clearly show that flies form a context dependent LTM (cLTM) after a single training trial. I think this alone is worthy of publication.

Surprisingly the authors show that this cLTM cannot be disrupted by cycloheximide feeding, or cold-shock applied immediately after training. These data are certainly suggestive that cLTM is quite unique and so they are important. They are however, rather traditional crude and don't allude to any detailed mechanism.

The authors do a few experiments to investigate the underlying neural mechanisms. They show that cLTM requires TH-Gal4 labelled neurons during training. This is a good start but obviously TH-GAL4 labels many neurons that have previously been implicated in aversive learning. This literature however focuses on the mushroom body (MB). Zhao et al contend that cLTM is independent of the MB because the DA receptor is not required in MB neurons and they can block MB output during retrieval, without consequence. Again, these are really interesting results that imply that DA signals to elsewhere are critical. They do not investigate this further though.

Instead the authors follow up by showing that retrieval requires olfactory projection neurons. This seems obvious because it's hard to imagine how the odors would otherwise be recognized and they are presumably still the instructive cue even for cLTM. Other experiments show that some neurons in the lateral horn are also required to express cLTM. Again, these are interesting, although they seem rather preliminary.

So, in summary, I find this an interesting short story with some good experiments. The results are worth publishing in their own right, but the current understanding of mechanism is rudimentary.

Reply #1:

We thank the reviewer for the positive comment as that “*The results are worth publishing in their own right*” and share the view as that “*the current understanding of mechanism is rudimentary*”. We are also interested in mechanisms underlying this unexpected finding, but such efforts are beyond the scope of this manuscript, which is of showing that an LTM is formed without requirement of consolidation. We will further investigate the molecular and neural circuitry mechanisms associated with cLTM in the future.

Reviewer #2:

Major-1:

The introduction moves from flies to humans and back to flies, and could be refocused to setup the significance of the results section. Also, there is discussion of human contextual memory, but extensive work in rodents is not mentioned. It would be very helpful to add discussion of this alongside work in humans.

Reply #2.1:

We thank the reviewer for this suggestion. In revision, we add discussion of context-dependent memory in rodents (line 53-55). We would like to point out that context-dependent memory is different from the extensively investigated contextual memory. For contextual memory, the directly fear response to the context is measured, while context-dependent memory refers to the phenomenon that the memory to a stimulus (odor or sound) is retrieved better in the encoding context.

Major-2:

In figure 1C it would be useful to provide competing contexts. Trained odor with copper-grid vs. untrained odor with copper grid.

Reply #2.2:

We have performed the suggested experiments. Trained flies tend to avoid trained odor, which was paired with electric shock, and choose the untrained odor (new odor). Consistent with our findings, this trend is intensified in the copper grid (Supplementary Fig.1b, revised manuscript). Besides, we also confirmed that memory in untrained flies cannot be increased with presence of the copper grid (Supplementary Fig.1a, revised manuscript).

Major-3:

Were flies outcrossed into the control w1118 strain to control for background?

Reply #2.3:

Relevant information is included in revision (line 380-386). The following strains used for the memory experiments were outcrossed to the w1118 background: *TH-Gal4*, *OK107-Gal4*, *C772-Gal4*, *nSyb-Gal4*, *OK66-Gal4*, *GHI46-Gal4*, *MZ699-Gal4*, *R38E07-Gal4*, *GMR-Gal4*, *UAS-Shibire^{ts1}*, *UAS-dDA1-RNAi*. However, the following strains used for the memory experiments were not outcrossed, including *NP1004-Gal4*, *MZ671-Gal4*, *NP3060-Gal4*, *NP5194-Gal4*, *NP2492-Gal4*, because all these strains are used for comparison only in the context of the same genotype but at different conditions. Therefore, the genetic background of these flies should not affect the outcome of comparison.

Major-4:

I do not like the use of 'Life-long memory.' Indeed 15 days is a long time for the fly, but it is not lifelong and it would be better to say something like protein synthesis-independent long-term memory.

Reply #2.4:

As the reviewer suggested, we removed “life-long” in title and in the text in revision. The cLTM is described as protein synthesis-independent long-term memory.

Major-5:

The characterization of the sensory processes involved seems insufficient and jumps to central brain circuitry without a detailed characterization of sensory inputs. Were training and testing performed in the light or dark? What happens to vision mutants? mechanosensory mutants? The model in Figure 6 describes different inputs but these are not directly tested.

Reply #2.5:

First, all behavioral assays were performed in dark room with dim red-light source (as described in Tully and Quinn, 1985). Second, additional sensory-relevant manipulation was performed and data are included in revision. Considering potential developmental effects may result from the use of mutants, we used acutely inducible manipulation through UAS-*Shibire^{ts}* to block eyes and optic lobe specifically during cLTM retrieval (line 243-248 in revision). Results show the visual system is required for cLTM retrieval (Supplementary Fig. 6a). For the mechanosensory input, the related investigation is included in the old manuscript as that, removing arista, the major mechanosensory organ, after training would completely block cLTM retrieval (Fig. 4d). These data support the model in Fig. 6 that multiple inputs are required to retrieve cLTM. This is clarified further in revision (line 219-221).

Major-6:

While the paper clearly shows cLTM is different from standard LTM, it would seem important to test canonical cAMP signaling pathways to see if these are shared.

Reply #2.6:

We have performed the suggested experiments with two different classical mutations of cAMP signaling pathway, *rut¹* and *rut²⁰⁸⁰*. The obtained data shows cLTM formation is not dependent on cAMP pathway (Fig. 1h).

Major-7:

My understanding is LTM is formed in a single trial using olfactory taste memory and mammalian shock memory. This may be important to reference.

Reply #2.7:

We mentioned olfactory memory (appetitive memory) in old manuscript. We now stress this point with vertebrate contextual memory together in revision, as suggested (line 293-398).

Minor-1:

The opening sentence (Line 25) seems subjective. It may be better to biologically define 'difficult to form'.

Reply #2.8:

As reviewer suggested, we rephrased the thirteenth two sentences for making it more biological (line 26-28).

Minor-2:

Line 30, include 'spaced'?

Minor-3:

Line 64: 'innate' instead of 'memory-irrelevant'?

Reply #2.9:

We thank the reviewer for these helpful suggestions and modified the text accordingly.

Minor-4:

Line 367. Please describe the control solution for CXM.

Reply #2.10:

We appreciate that the reviewer pointed out our mistake. We add the description of the control solution for CXM in *Methods* (line 419-420).

Reviewer #3:

Major-1:

Another protein synthesis inhibition and dose dependency of cycloheximide for cLTM formation: Authors only test cycloheximide without checking its dose dependency. Since this is a major claim of this paper, reviewer want to know whether other protein synthesis inhibition still do not affect cLTM formation. For example, RICIN employed by Chen et al Science (2012), is a gene encoding inhibitor of protein synthesis.

Reply #3.1:

As suggested by the reviewer, we perform new experiment with pan-neuronal expressed RICIN (*nSyb-Gal4;UAS-RICIN*) in revision (Supplementary Fig. 1g). As described by *Chen et al Science (2012)*, we move flies to high-temperature (30°C) immediately after training until test. The new result is consistent with our finding in the initial submission with CXM that inhibiting protein synthesis do not affect cLTM formation.

Major-2:

LH>shi(ts) experiments; Since authors claim that LH is a center of cLTM, reviewer need to know whether inhibiting outputs from LH affect cLTM retrieval. Also, if LH is

exclusively involved in cLTM (as shown in Figure 6), LH>shi(ts) should have no effects on classical LTM.

Reply #3.2:

We have performed the suggested experiments (line 266-274). The obtained data support that inhibiting outputs from LH affect cLTM retrieval (Supplementary Fig. 5c). We included different types of LH output neurons labeled by six different Gal4s, and we found that five of them are required for cLTM retrieval. Mushroom body output neuron, MB-V2, to LH is reported to affect classic LTM retrieval, as referred in manuscript (line 262). As suggested, we performed additional experiments to test effects of inhibiting other LH neurons during classical LTM retrieval (Supplementary Fig. 5b). The result shows that LH neurons labeled by *NP1004-Gal4*, *MZ671-Gal4*, *NP3060-Gal4*, and *NP5194-Gal4* are not involved in traditional LTM retrieval. These data support that LH is a center of cLTM.

Major-3:

cLTM is a LTM?: Given that LTM is defined that it is depending on gene expression and protein synthesis, cLTM may be another form of anesthesia-resistant memory (ARM). Hence, it is worth to check cLTM in ARM mutants, such as radish.

Reply #3.3:

We thank the reviewer for this suggestion and provide new data in revision that *radish* performs normal cLTM (Supplementary Fig. 1h). This data supports that cLTM is not another form of ARM.

Minor-1:

Reviewer curious to know if cLTM is also independent of transcription.

Reply #3.4:

Considering the critical role of transcription factor CREB in memory, we provide data that the classical CREB-dependent transcription is not required for cLTM (Fig. 1g).

Minor-2:

Among various contexts including vision, tactile, temperature, why only odor can form context independent memory?

Reply #3.5:

We completely agree with this concern. In fact, various context elements, including vision, tactile, temperature, were all presented in the whole training process (5min), so we call them context information. However, the conditioned odor (CS) was given with electric shock (1min), so it is the only element paired with shock rather than the others, and forms associative memory.

Minor-3:

Which brain regions require D1 function for cLTM formation?

Minor-4:

Further mechanical insights into cLTM?

Reply #3.6:

We have the similar interest in the question that which brain regions require D1 function for cLTM formation. Indeed, we have made some preliminary explorations of this question. So far, we have not found this region. As reviewer suggested, we add related discussion in the revised *Discussion* section (line 350-356). And thus, it would provide further mechanical insights to cLTM.

Reviewers' Comments:

Reviewer #1:

Remarks to the Author:

I remain convinced that this is an interesting manuscript with some key results. This is despite the authors seeming to have taken my prior enthusiasm as a sign that they could overlook all of my criticism, as 'beyond the scope of this manuscript'.

I do not think their current explanation for the phenomenon being MB independent makes sense, but they are the authors.

I have no further comment to make.

Reviewer #2:

Remarks to the Author:

In the revised version the authors have been highly responsive to initial comments and addressed all significant concerns. This manuscript challenges a central dogma in the field, and will have a major impact. It is still not clear to me why contextual memory would not require the molecular and neural machinery of non-contextual memory, in addition to other sensory modalities, and instead occurs through an entirely new mechanism. However, the model provided is fully supported by the data presented.

Line 344 *long been seemed

Reviewer #3:

Remarks to the Author:

Authors appropriately addressed my concerns.